# Genetic characterization of a mild isolate of papaya ringspot virus type-P (PRSV-P) and assessment of its cross-protection potential under greenhouse and field conditions

**Andres X. Medina-Salguero**[1☯], **Juan F. Cornejo-Franco**[1☯], **Sam Grinstead**[2], **Joseph Mowery**[3], **Dimitre Mollov**[2], **Diego F. Quito-Avila**[1,4]*

**1** Centro de Investigaciones Biotecnológicas del Ecuador, Escuela Superior Politécnica del Litoral, ESPOL, Guayaquil, Guayas, Ecuador, **2** National Germplasm Resources Laboratory, USDA-ARS, Beltsville, MD, United States of America, **3** Electron and Confocal Microscopy Unit, USDA ARS, Beltsville, MD, United States of America, **4** Facultad de Ciencias de la Vida, Escuela Superior Politécnica del Litoral, ESPOL, Guayaquil, Guayas, Ecuador

☯ These authors contributed equally to this work.
* dquito@espol.edu.ec

**Data Availability Statement:** All relevant data are within the paper and its Supporting Information

## Abstract

A mild isolate of *Papaya ringspot virus* type-P, abbreviated as PRSV-mild, from Ecuador was sequenced and characterized. The most distinguishing symptom induced by PRSV-mild was gray powder-like leaf patches radiating from secondary veins. In greenhouse experiments, PRSV-mild did not confer durable protection against a severe isolate of the virus (PRSV-sev), obtained from the same field. Furthermore, isolate specific detection in mixed-infected plants showed that PRSV-sev becomes dominant in infections, rendering PRSV-mild undetectable at 90–120 days post superinfection. Virus testing using isolate-specific primers detected PRSV-mild in two out of five surveyed provinces, with 10% and 48% of incidence in Santo Domingo and Los Ríos, respectively. Comparative genomics showed that PRSV-mild lacks two amino acids from the coat protein region, whereas amino acid determinants for asymptomatic phenotypes were not identified. Recombination events were not predicted in the genomes of the Ecuadorean isolates. Phylogenetic analyses placed both PRSV-mild and PRSV-sev in a clade that includes an additional PRSV isolate from Ecuador and others from South America.

## Introduction

*Papaya ringspot virus* (PRSV) is a positive-sense single-stranded RNA virus in the genus *Potyvirus*, family *Potyviridae*. Based on host range, PRSV is grouped into two serologically indistinguishable biotypes: PRSV type-P (PRSV-P) and PRSV type-W (PRSV-W) [1]. While PRSV-P isolates can infect species in the families *Caricaceae*, *Cucurbitacea* and *Chenopodiacea*, isolates of the W type infect only species in the *Cucurbitacea* and *Chenopodiacea* [1–3]. However, a PRSV report from *Robinia pseudoacacia* L., a tree species in the family *Fabaceae*, suggested a wider host range for the virus [4].

files. Sequences are available in NCBI under accession numbers: MT747167 MH974110.

**Funding:** This work was funded partially by the Ecuadorean University Network (REDU) project PREDU-2016-001, through ESPOL.

**Competing interests:** The authors have declared that no competing interests exist.

PRSV-P is the causal agent of papaya ringspot disease (PRD), one of the most destructive and economically important diseases of papaya worldwide [3]. The virus is transmitted in a non-persistent manner by several aphid species commonly found in weeds associated with papaya orchards [5–7].

Symptoms were first observed and described in 1949 in Hawaii [8], where the presence of concentric rings on the fruit and oily streaks on petioles and young parts of the trunk was characteristic of the disease. However, infected trees display an array of leaf symptoms from vein clearing and chlorosis to leaf deformation, defoliation and death [2,3].

The PRSV genome is ~10,320 nucleotides plus a poly(A) tract at the 3' end. The genome contains a single open reading frame (ORF), typically from nucleotides 86 to 10,120 encoding a polyprotein precursor of 3,344 amino acids, which is auto- and trans-proteolytically processed into ten mature proteins [3,9,10].

The genetics and diversity of PRSV have been studied extensively. Studies based on the coat protein (CP) suggested that PRSV originated from southern Asia, as higher diversity levels were found among Indian and other Asian isolates compared to sequences from the rest of the world [11–14]. Further analyses using complete genomes of isolates from different geographical regions showed that recombination events, especially at the 5' end, and P1 gene, have played a pivotal role in the diversification of PRSV, supporting previous findings on the origin of the virus [14,15].

Additional studies conducted on PRSV and closely related potyviruses have helped identify the genetic determinants for host specificity (P-type vs W-type), aphid transmission and symptom development [11,13,16–20]. The latter is particularly important for cross-protection—the use of a mild strain to confer protection against a severe strain of the same virus [21]. This management strategy has been explored for PRSV, although results have not been successful [18,22–24].

Nevertheless, in Ecuador, where PRSV management relies upon intensive roguing and vector control, the use of cross-protection may offer an opportunity for growers to reduce field losses due to this devastating disease. Therefore, the objectives of this study were to characterize a mild isolate of PRSV-P obtained from a commercial papaya planting of Ecuador and test its ability to confer cross-protection against a severe isolate from the same field.

## Materials and methods

### Ethics statement

Plant samples for this study were collected in accordance to Ecuadorean Government regulations under field permit number MAE–DNB–CM–2018–0098 granted by the Department of Biodiversity of the Ecuadorean Ministry of the Environment.

### Virus source and testing

In 2018, commercial papaya plants (cv. Tainung) showing leaf vein clearing and deformation, suggestive of PRSV, were observed next to plants showing mild leaf mosaic. Leaf samples from symptomatic and mildly affected plants were collected for testing by ELISA (Agdia, Inc.) and reverse-transcription (RT)-PCR from total RNA as described [25]. RT was done in a 15 μl mixture containing 150 ng of random primers, 100–150 ng of RNA, 1× first-strand buffer, 1 mM dNTPs, 10 mM dithiothreitol (DTT), and 60 U of reverse transcriptase RevertAid® (Thermo-Scientific). PCR was carried out in a mixture containing 1 × PCR buffer, 40 ng of primers PRSV_Ec-F: GAGARGTAYATGCCGCGGTATGG and PRSV_Ec-R: CGCATACCCAGGAGAGA GTGC, 2 mM dNTPs, 1.5 μl cDNA template, 0.1 μl of Taq DNA polymerase (Genescript, USA)

and water to a final volume of 10 ul. PCR parameters were: 94˚C for 4 min, 40 cycles of 94˚C for 45 s, 57˚C for 30 s and 72˚C for 45 s, and a final extension step of 10 min at 72˚C.

## Mechanical inoculation and host range

Two-month old papaya seedlings of two cultivars 'Tainung' and 'Sunrise' were used for mechanical inoculations following standard protocols [26]. For inoculum preparation, 500 mg of fresh infected tissue were ground and homogenized in 5 ml phosphate buffer 0.05 M, pH 7.0 using a mortar and pestle. Two fully developed young leaves were dusted with silicon carbide (mesh 320) and rubbed with the homogenate using a soft sponge.

For each cultivar, five plants were inoculated with the mild isolate of PRSV and another group of five plants were inoculated with the severe isolate of PRSV. Inoculated plants were kept in an insect-proof greenhouse and monitored for symptom development. PRSV infection was confirmed at 30 days post inoculation (dpi) by RT-PCR testing the youngest fully developed leaf from each inoculated plant.

In addition, an isolate from one of the inoculated papaya seedlings was used as virus source to inoculate, mechanically, three cucurbitaceous species: melon ('Expedition'), watermelon ('Ghalina') and cucumber ('Jaguar').

## Genome sequencing

The genomic sequences of both the severe and mild isolate were determined using high-throughput sequencing (HTS). Total RNA was extracted using a Qiagen RNA plant mini kit (Germantown, MD), treated with RNase-free DNase (Ambion) and subjected to ribosomal RNA depletion before TruSeq RNA library preparation [27]. HTS was performed on a Next-Seq 500 Illumina platform as single 75 bp reads.

Sequence reads were initially processed and assembled into contigs by CLC Workbench 11 (Qiagen USA). Genome assemblies were determined with Spades and Geneious 11 (Biomatters, New Zealand). Terminal sequences were obtained using 5' RACE Kit (ThermoFisher) and an anchored oligo-dT primer for the 3'end. Primer design, genome assembly and open reading frame (ORF) prediction were conducted using Geneious Prime (Biomatters, New Zealand).

## Phylogenetic analysis

Full length genome PRSV sequences (67 isolates) were downloaded from NCBI GenBank and phylogenetically compared with the two isolates from this study. Multiple sequence alignment (msa) was done using Clustal W in MegaX [28].

Recombination analysis was performed using the RDP4 package [29]. Recombinant regions were removed from the alignment before the phylogenetic inference, which was done using the Maximum Likelihood (ML) method with the Tamura-Nei model and bootstrap of 1,000 replicates in MegaX [28].

## Transmission Electron Microscopy (TEM)

Fresh young leaves were detached from a plant infected with the mild or severe isolate and cut into 1 mm disks using a biopsy punch. Leaf disks were submerged in a fixative solution containing 2% paraformaldehyde, 2.5% glutaraldehyde, 0.05% Tween-20 and 0.05M sodium cacodylate and 0.005M calcium chloride and processed in a Pelco BioWave microwave as previously described [30]. Ultrathin sections were cut at 60 nm, with a Leica UC7 ultramicrotome with a Diatome diamond knife and mounted onto 100 mesh carbon/formvar coated

copper grids. TEM grids were stained with 3% Reynolds lead citrate for 5 min and imaged at 80 kV with a Hitachi HT-7700 transmission electron microscope (Hitachi High Tech America, Inc., Dallas, TX, USA).

## Differential detection and field surveys

Sequence alignments were used to design primers capable of discriminating between the severe and mild PRSV isolates. To that end, several primers were evaluated in a duplex PCR format after reverse-transcription using random primers. The sensitivity of selected primers was assessed using equal number of *in vitro* generated transcripts in ten-fold serial dilutions and resuspended in PRSV-free papaya RNA (100 ng/μl).

The RT-PCR detection assay was used to monitor isolate interactions during cross-protection experiments, and for surveying the incidence of both isolates in a total of 340 samples from commercial orchards of five papaya-producing provinces of Ecuador: Los Ríos, Santo Domingo, Santa Elena, Manabí, and Guayas. Two-year-old plantings of 'Sunrise' and 'Tainung', currently the two most cultivated papaya varieties, were selected for the survey. At each field, the youngest fully developed leaves were collected, with plants sampled individually in a zigzag pattern. Symptoms of sampled leaves were recorded in the laboratory.

## Cross-protection under greenhouse conditions

The capacity of a mild isolate of PRSV to protect papaya plants from a secondary infection of a severe isolate was tested under greenhouse conditions. Mechanical inoculations were done as described above (see: mechanical inoculation and host range).

Leaves from plants infected with the mild isolate were used to inoculate a set of five 'Tainung' and five 'Sunrise' two-month-old papaya seedlings (immunization step). A mock-immunized plant was added as a control for each cultivar. Virus infection was confirmed by RT-PCR on newly developed leaves at 30 days post immunization (dpim), upon which the severe isolate was inoculated (superinfection step) on each immunized or mock-immunized plant.

Plants were maintained in an insect-proof greenhouse under standard conditions (12 h light and 25 C) and monitored for symptoms. At 30, 60, 90 and 120-days post superinfection (dpsi), plants were tested for both isolates as described above, and symptoms were monitored for six months under greenhouse conditions.

## Sentinel plants

In order to monitor the spread and interactions of both PRSV isolates under field conditions, a set of twelve PRSV-free "sentinel" papaya plants ('Tainung') were planted in the same commercial field of Los Ríos province, from where both PRSV isolates were first obtained.

Sentinels were scattered in an area of approximately 300 m$^2$ of a two-year-old lot and tested for both isolates on a monthly basis starting from 90 days post planting (dpp). Testing was done on approximately 100 mg of the youngest fully developed leaf using the duplex RT-PCR assay developed in this study. Symptoms on leaves, stem and fruits were recorded at each sampling time.

# Results

## Two distinct PRSV isolates found in papaya plants showing severe and mild symptoms

Mechanically inoculated papaya plants tested positive for PRSV at 30 dpi. Plants inoculated with extracts from mildly affected leaves displayed subtle gray powder-like patches, which radiated from the secondary veins of young fully-developed leaves; whereas plants inoculated

with the extracts from severely affected leaves showed vein clearing, typical oily streaks on the stems and stunting (Fig 1). During the course of this study, however, symptoms induced by the mild field isolate on inoculated plants were intermittent ranging from asymptomatic to moderate leaf mosaic (conspicuous mosaic with vein clearing) phases. Such variation in symptom expression was not observed in papaya plants infected with the severe field isolate.

Both isolates were detected by RT-PCR in mechanically inoculated cucurbitaceous species. No symptoms were observed in plants infected with the mild isolate, whereas vein clearing and leaf mosaic were observed in those infected with the severe isolate. Based on the distinct symptoms induced by each PRSV isolate, the terms PRSV-mild and PRSV-sev will be used to refer to the mild and severe isolates, respectively.

## Genome and phylogeny

The PRSV-mild sample produced 14,950,447 HTS data reads which assembled to approximately 40,000 contigs. Five contigs were identified with high identities to PRSV. Over three million reads were mapped to these contigs and used to assemble the genome. After obtaining the terminal reads, PRSV-mild genome assembled as 10,320 nucleotide (nt) long excluding the Poly (A) tract. The average coverage per nucleotide position was 22,275 x. The single open reading frame (ORF) was located at nt positions 86–10,114. The PRSV-mild genomic sequence was deposited in NCBI under accession number MT747167.

The PRSV-sev sample generated 32,375,528 total HTS reads yielding 61,000 contigs, of which 27 were similar to PRSV. Out of all reads, 3,846,945 (~11.9%) were mapped to PRSV

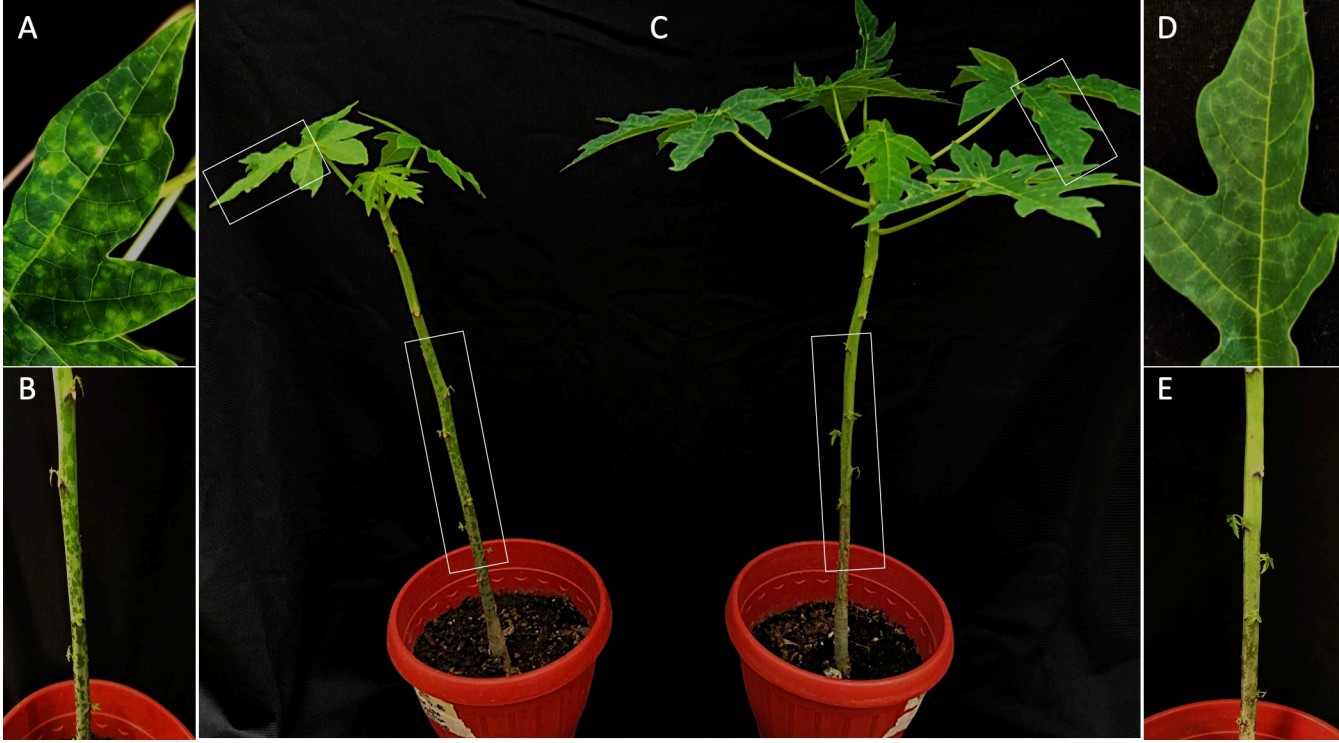

**Fig 1. Symptom expression in papaya plants infected with papaya ringspot virus (PRSV).** Severe leaf vein clearing (A) and oily streaks on the stem (B) induced by the severe isolate of PRSV (PRSV-sev). C) Overall growth comparison between plants infected with PRSV-sev (left) and the mild isolate (PRSV-mild) (right). Mild powder-like patches radiating from leaf secondary veins (D) and absence of oily streaks (E) induced by PRSV-mild. Specific symptoms on leaves and stems are shown by white boxes.

contigs. Subsampling from these reads to 30,000 yielded a single contig assembly of the PRSV genome of initial length 10,298 nts. After terminal reactions, the final assembly was determined to be 10,324 nt long excluding the Poly (A) tail. The single ORF spans from nt positions 86–10,120. The PRSV-sev genomic sequence was deposited in NCBI under accession number MH974110.

Sequence comparisons between the two isolates showed 95.4% and 96% identity at the nucleotide and amino acid level, respectively. The terminal regions 5'-AAATAAAACATCT and -CTCTTAGAATGAG-3' were conserved in both isolates. Similar sequence identities (95–96%) were observed between the newly sequenced genomes and a PRSV isolate (MH974109) from a mountain papaya species, known as babaco (*Vasconcellea x heilbornii*), recently reported from Ecuador [31].

More than 20 recombination events were identified in 18 PRSV isolates mostly from Asia, but not in Ecuadorean isolates (not shown). This result was consistent with findings from a recent study, which showed a similar number of recombination events concentrated at the 5' terminus and the virus protein 1 (P1) [14]. The phylogenetic reconstruction, excluding the recombinant regions in Asian isolates, showed that the Ecuadorean PRSV genomes are related more closely to Colombian isolates in a clade that contains sequences from the United States, Mexico and France (Fig 2).

Analysis of the polyprotein showed the presence of typical RNA-dependent-RNA-polymerase (RdRp) conserved motifs QPSTVVDN and GDD at aa positions 2,831–2,938 and 2,870–2,872, respectively, within the nuclear inclusion b (NIb) region. The presence of the K residue, located at aa position 2,039 in the nuclear inclusion a protease (NIaPro) region, which is considered a papaya host determinant [19] supported the P-type nature of both PRSV-mild and PRSV-sev.

The putative aphid transmission determinants KITC (polyprotein aa position 598) and PTK (aa 856) in the HC-Pro; and the DAG, WCIEN and QMKAAA motifs in the coat protein (CP) region (aa positions 3,064, 3,194 and 3,295, respectively) [32,33] were conserved in both PRSV isolates. The HC-Pro motif FRNK(X)$_{12}$CDN has been suggested as a symptom determinant in members of the *Potyvirus* genus, as experimentally shown for *zucchini yellow mosaic virus* (ZYMV) [16,17]. In PRSV-mild and PRSV-sev a FRN**K**-to-FRN**R** variation was observed in such motif. Interestingly, an identical variation was also observed in a Venezuelan isolate of another potyvirus *Zucchini tigre mosaic virus*, which induces severe leaf chlorotic stripes on zucchini [34]; suggesting that the K-to-R change in the above mentioned motif does not abolish severe symptom expression.

Lastly, a 6-nt deletion was observed at the amino-terminus of the CP in PRSV-mild (genome position 9,348) but not in PRSV-sev. Genome comparisons revealed that such a deletion was also present in an isolate from USA-Hawaii (EU126128), USA-Texas (KY271954), India (EF017707) and Mexico (AY231130), and resulted in the loss of residues KK/E (S1 Fig).

## Transmission Electron Microscopy (TEM)

Large quantities of pinwheel inclusions and aggregations of filamentous virus particles were observed in the leaf samples infected with PRSV-sev but not in the samples infected with PRSV-mild (S2 Fig).

## Virus detection and surveys

PRSV-mild was specifically detected by primers F: 5-GGAGATCAACTCCGTCGATTTCAAT-3 and R: 5-GTCTACTCCTCTTATAACATTGCCG-3, which flank a 647 bp fragment of the P1 gene; whereas, PRSV-sev was specifically detected by primers F: 5-TAAATACTTTGAGCGTG

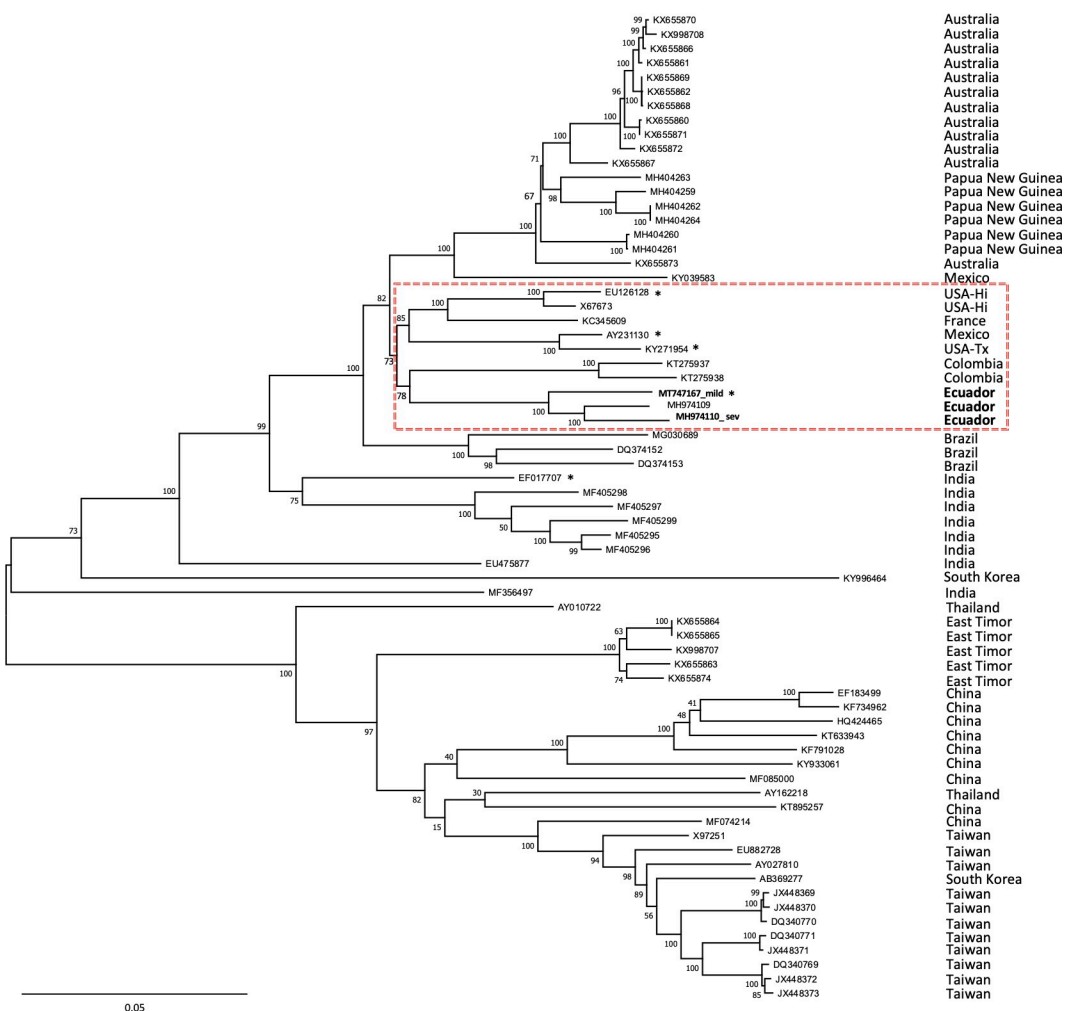

**Fig 2. Phylogenetic reconstruction of papaya ringspot virus (PRSV).** Sixty-seven PRSV complete genomes were aligned and used for phylogenetic inference using the Maximum Likelihood method with the Tamura-Nei model and bootstrap of 1,000 replicates (shown at each node). Taxa are indicated by accession number. The country where each isolate was reported is shown on the right. Dashed red box indicates a subclade including isolates from Ecuador, Colombia, United States, France and Mexico. Asterisk denotes genomes with the six-nucleotide deletion in the EK region of the coat protein gene.

AGAGGGGA-3 and R: 5-CACTCCCTCATACCACTTCTCAAAT-3, which amplify a 469 bp fragment of the CP gene. Primer sensitivity assays showed a comparable detection limit of ~ 300 copies of target for both primer sets in single-or duplex RT-PCR (S3 Fig). Both isolates were indistinguishably detected by PRSV ELISA test (Agdia, Inc.).

Virus surveys indicated that PRSV-mild was prevalent in Los Ríos, with 46 and 48% of positives in 'Sunrise' and 'Tainung', respectively; whereas the severe isolate was found in 20 and 25% of samples. There were no samples positive for both isolates. Leaf symptoms observed in plants infected with PRSV-mild ranged from mild to moderate mosaic; whereas those infected with the severe isolate displayed severe vein clearing and leaf deformation. In Santo Domingo, only 10% of the analyzed 'Sunrise' plants tested positive for PRSV-mild. Symptoms observed in PRSV-mild positive plants consisted of mild leaf mosaic. The severe isolate was not detected in this province. In Guayas, Manabí and Santa Elena, PRSV-mild was not detected; whereas the incidence of PRSV-sev ranged from 50 to 100% (Table 1).

**Table 1. Field occurrence of the mild and severe isolates of papaya ringspot virus (PRSV-mild and PRSV-sev).**

| Province | Cultivar | Sampling size (n) | Isolate occurrence (%) | |
|---|---|---|---|---|
| | | | PRSV-mild | PRSV-sev |
| Los Ríos[#] | Sun. | 50 | 46 | 20 |
| | Tai. | 50 | 48 | 25 |
| Santo Domingo | Sun. | 30 | 10 | 0 |
| | Tai. | 30 | 0 | 0 |
| Santa Elena | Sun. | 30 | 0 | 53 |
| | Tai. | 30 | 0 | 54 |
| Guayas | Sun. | 30 | 0 | 50 |
| | Tai. | 30 | 0 | 60 |
| Manabí | Sun. | 30 | 0 | 100 |
| | Tai. | 30 | 0 | 100 |

Two cultivars: Sunrise (Sun.) and Tainung (Tai.) were sampled at each of five provinces. # no mixed infected plants were observed during the survey.

### Cross-protection under greenhouse conditions

Plants immunized with PRSV-mild tested negative for PRSV-sev and showed only mild powder-like patches on upper non-inoculated leaves at 30 dpsi; whereas mock immunized plants tested positive for PRSV-sev at 30 dpsi and developed typical vein clearing and blistering by 45–60 dpsi (Fig 3). The 'protecting' effect, however, was not sustained over time. Immunized plants developed symptoms similar to those observed in mock-immunized plants between 90 and 120 dpsi. Interestingly, RT-PCR testing revealed that plants, which were no longer protected, tested positive for the severe isolate only; whereas the mild isolate could not be detected for the rest of the study.

### Occurrence of both PRSV isolates under field conditions

The occurrence of both PRSV isolates was monitored in sentinel plants. At 90 dpp, nine of the twelve plants were found infected. Five were positive for PRSV-mild only, three tested positive for PRSV-sev only, and one was co-infected with both isolates. At 150 dpp, the three remaining plants were also found infected. While most plants infected with either PRSV-sev or PRSV-mild remained singly infected during the course of the study, two plants that were found co-infected with both isolates initially became dominated by PRSV-sev single infections by 120 and 180 dpp, respectively, and remained that way (Fig 4).

Symptom expression on newly developed leaves and fruits varied depending on the type of infection in each sentinel plant. Plants infected with PRSV-sev, consistently showed severe leaf vein clearing and fruit deformation, and ultimately plant death by 240 dpp. Plants infected with PRSV-mild displayed intermittent symptoms from mild to severe, including ringspots on the fruits, with only one plant remaining completely asymptomatic during the course of the study (Table 2).

### Discussion

In this study, we characterized a mild isolate of PRSV- type P (here abbreviated PRSV-mild), found in a commercial papaya planting of Los Ríos province of Ecuador, and investigated its potential use in cross-protection against a typical severe isolate (PRSV-sev) obtained from the same field.

Mechanical inoculations onto two-month old papaya seedlings demonstrated that PRSV-mild induced gray powder-like patches on both cultivars used in this study. However,

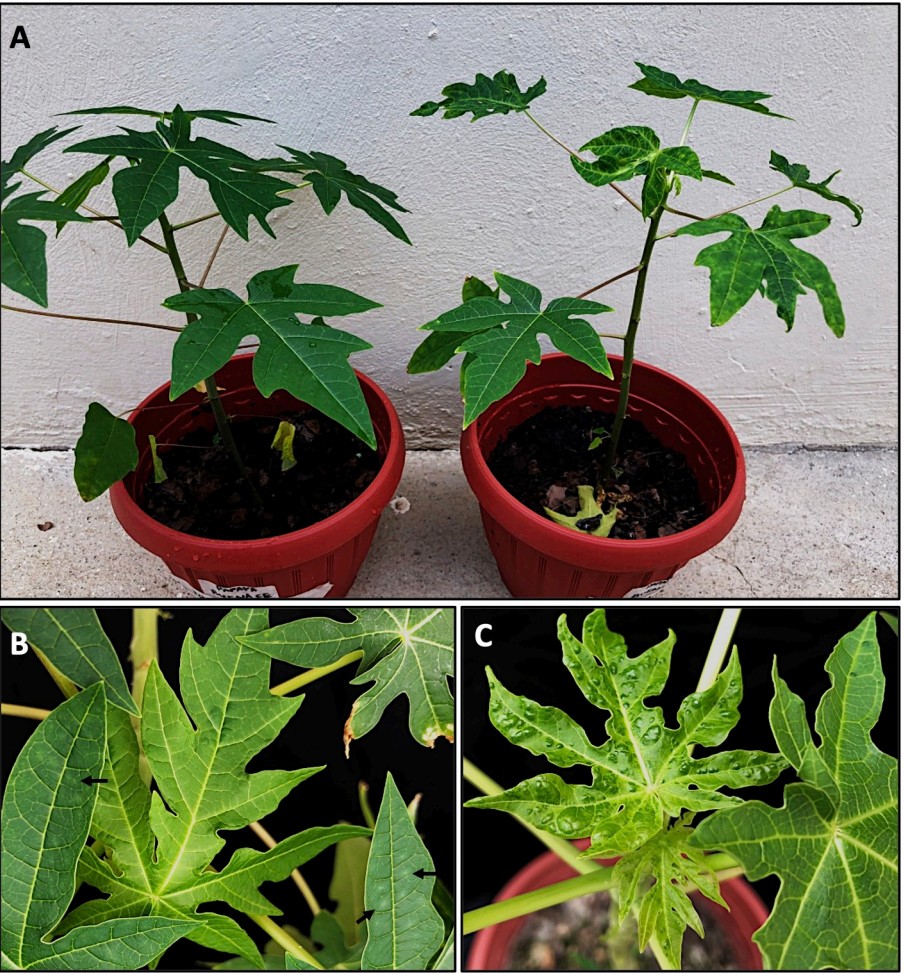

**Fig 3. Cross-protection effect of a mild isolate of papaya ringspot virus (PRSV-mild) against a severe isolate (PRSV-sev).** A) Overall growth comparison between an immunized (PRSV-mild infected) and a mock-immunized plant at 45 days post superinfection. B) Mild leaf symptoms on immunized plant. Arrows show the mild powder-like patches radiating from secondary veins. C) Severe blistering observed in the mock-immunized plant induced by PRSV-sev.

symptom expression was intermittent during the study, going from asymptomatic to moderate leaf mosaic phases. Conversely, PRSV-sev induced persistent severe vein clearing, leaf deformation and oily streaks on the stems (Fig 1).

The genome identity between the two isolates was 95%, clustering in a phylogenetic clade with a recently sequenced Ecuadorean PRSV isolate from babaco, a mountain papaya relative (*Vasconcellea x heilbornii*), and two Colombian sequences. This clade shares a most recent ancestor with isolates from the USA (Hawaii and Texas), Mexico and France (Fig 2), supporting findings from other studies where a correlation between sequence diversity and geographic area was not apparent [11]. No recombination events were predicted for any of the Ecuadorean or other American isolates, whereas the majority of recombinants were detected in Asian isolates, supporting previous results that place Asia as the origin center of PRSV [13,15,35].

An unusual 6-nt deletion was observed in the CP region of PRSV-mild. A similar deletion was found in an isolate from USA-Hawaii (EU126128), USA-Texas (KY271954), Mexico (AY231130) and India (EF017707), suggesting that this genomic feature may have originated

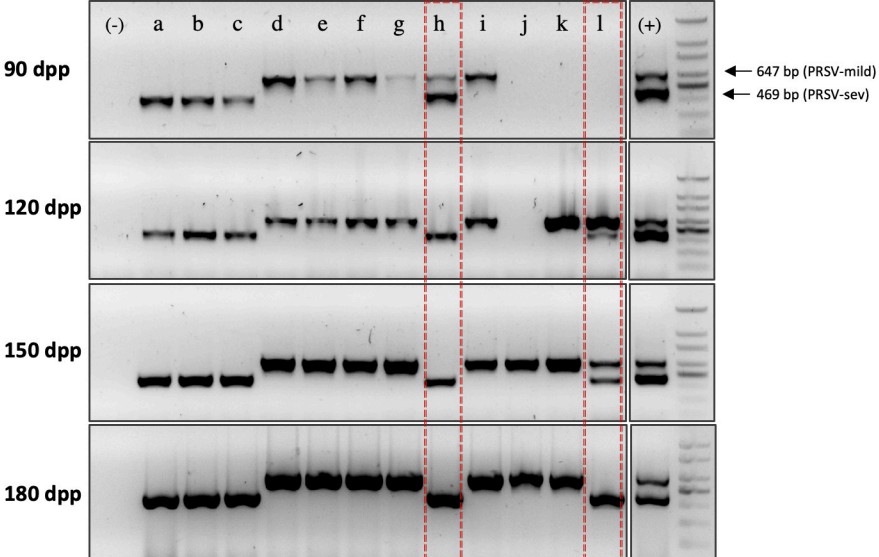

**Fig 4. Dynamics of the papaya ringspot virus mild (PRSV-mild) and severe (PRSV-sev) isolates under field conditions.** Progressive testing of PRSV-mild and PRSV-sev by reverse-transcription PCR. Different testing time points are indicated (dpp: Days post planting). Arrows next to the DNA ladder indicate the amplification size (base pairs, bp) for each isolate. A set of 12 sentinel plants (represented by letters a through l) were exposed to virus infection. Dashed red boxes denote plants where co-infection of both strains was observed, which then turned into PRSV-sev single infections.

and introduced from Asia. Considering that all isolates from a same geographical region e.g. Ecuador, Mexico or Hawaii, did not have the deletion, it is reasonable to speculate that more than one PRSV introduction events occurred from Asia to the Americas. However, a greater number of PRSV complete genome sequences from American isolates are needed to explain its continental introduction and evolutionary pathway.

The 6-nt deletion observed in PRSV-mild resulted in the loss of residues KK/E from the "EK region" located at the amino terminus of the CP. This repetitive EK region in PRSV was first observed by Silva-Rosales et al. [36] during a CP-based phylogenetic study using isolates from Mexico, Australia and Asia. Although the implication of this region in transmission and/ or virus replication was not determined, its location, downstream the DAG motif—an aphid transmission determinant [20]—suggested a potential role in transmission. In a recent poty-virus-wide genomics study, the CP N-terminal region surrounding the DAG motif was found to be hypervariable and suggested as a genetic feature for transmission adaptation to genetically diverse hosts and vectors [37]. Here, we observed that plant-to-plant transmission of PRSV-mild under field conditions was apparently not affected by the KK/E deletion. However, further experiments are needed to determine whether such a deletion plays a role in transmission efficiency by its natural vector.

The identification of pathogenicity factors, how these interact with host elements, and the potential use of mild strains in cross-protection against severe strains, are topics of intensive investigation for potyviruses [18,22–24,38]. In ZYMV, a widely spread potyvirus of cucurbit crops, virulence determinants were identified where amino acid changes: R to I and D to Y in the highly conserved HC-Pro motif F**R**NK(X)$_{12}$C**D**N, respectively, resulted in abolishment of leaf symptoms in their natural hosts [16,17]. In this study, we found that both PRSV-mild and-sev had a FRN**K**-to-FRN**R** variation in such a conserved motif. An identical amino acid change was observed in the babaco isolate of PRSV (from Ecuador) and a symptom-inducing

**Table 2. Dynamics of the mild and severe isolate of papaya ringspot virus (PRSV-mild and PRSV-sev) under field conditions.**

| | 90 dpp | | | | | 120 dpp | | | | | 150 dpp | | | | |
|---|---|---|---|---|---|---|---|---|---|---|---|---|---|---|---|
| | Symptoms | | | RT-PCR | | Symptoms | | | RT-PCR | | Symptoms | | | RT-PCR | |
| Sentinel | leaf | fruit | stem | PRSV-sev | PRSV-m | leaf | fruit | stem | PRSV-sev | PRSV-m | leaf | fruit | stem | PRSV-sev | PRSV-m |
| a | v.c. | n.a. | y | + | - | blist. | n.a. | y | + | - | v.c. | rings | y | + | - |
| b | mos. | n.a. | n | + | - | blist. | n.a. | y | + | - | v.c. | n.a. | y | + | - |
| c | v.c. | n.a. | y | + | - | blist. | n.a. | y | + | - | v.c. | n.a. | y | + | - |
| d | mos. | n.a. | y | - | + | mos. | n.a. | y | - | + | mos. | n.a. | y | - | + |
| e | mos. | n.a. | y | - | + | mos. | n.a. | n | - | + | asymp. | n.a. | n | - | + |
| f | mos. | n.a. | n | - | + | mos. | n.a. | n | - | + | v.c. | clean | n | - | + |
| g | mos. | n.a. | y | - | + | mos. | n.a. | y | - | + | v.c. | rings | y | - | + |
| h | mod. | n.a. | n | + | + | blist. | n.a. | y | + | - | v.c. | def. | n | + | - |
| i | mod. | n.a. | n | | + | blist. | n.a. | n | - | + | v.c. | n.a. | y | - | + |
| j | asymp. | n.a. | n | - | - | asymp. | n.a. | n | - | - | asymp. | n.a. | n | - | + |
| k | asymp. | n.a. | n | - | - | mos. | n.a. | n | - | + | mod. | n.a. | n | - | + |
| l | asymp. | n.a. | n | - | - | mos. | n.a. | n | + | + | v.c. | n.a. | y | + | + |

| | 180 dpp | | | | | 210 dpp | | | | | 240 dpp | | | | |
|---|---|---|---|---|---|---|---|---|---|---|---|---|---|---|---|
| | Symptoms | | | RT-PCR | | Symptoms | | | RT-PCR | | Symptoms | | | RT-PCR | |
| Sentinel | leaf | fruit | stem | PRSV-sev | PRSV-m | leaf | fruit | stem | PRSV-sev | PRSV-m | leaf | fruit | stem | PRSV-sev | PRSV-m |
| a | v.c. | def. | y | + | - | v.c. | def. | y | + | - | # | # | # | # | # |
| b | v.c. | def. | y | + | - | v.c. | def. | y | + | - | v.c. | def. | y | + | - |
| c | v.c. | rings | y | + | - | v.c. | rings | y | + | - | # | # | # | # | # |
| d | mod. | rings | y | - | + | blist. | rings | y | - | + | asymp. | rings | y | - | + |
| e | v.c. | rings | y | - | + | mos. | clean | y | - | + | mod. | clean | y | - | + |
| f | v.c. | rings | n | - | + | mos. | rings | n | - | + | asymp | clean | n | - | + |
| g | mod. | rings | y | - | + | v.c. | rings | y | - | + | blist. | n.f. | y | - | + |
| h | v.c. | n.f. | n | + | - | v.c. | n.f. | n | + | - | # | # | # | # | # |
| i | mod. | rings | y | - | + | mod. | rings | y | - | + | v.c. | rings | y | - | + |
| j | asymp. | clean | n | - | + | asymp. | clean | n | - | + | asymp. | clean | n | - | + |
| k | v.c. | n.f. | n | - | + | v.c. | rings | n | - | + | v.c. | clean | n | - | + |
| l | v.c. | def. | n | + | - | v.c. | def. | n | + | - | v.c. | rings | n | + | - |

Symptoms and infection status, tested by reverse-transcription PCR, were evaluated on a monthly basis starting at 90 days post planting (dpp) and symptom monitoring under field conditions. Sentinel plants are denoted by letters from a through l. Symptoms were monitored on leaves, fruits and stems. Symptom abbreviations: v.c.: Vein clearing; mos.: Mild mosaic; mod.: Moderate mosaic including mild vein clearing; blist.: Blistering and leaf deformation; asymp.: Asymptomatic; n.a.: Not applicable (trees are not in production stage); def.: Deformed fruits with rings; n.f.: Nonproductive tree (due to virus); clean: Fruit not showing rings; y: Presence of oily streaks; n: Absence of oily streaks; # dead plant. + indicates a positive result for the isolate;—denotes a negative result for the isolate.

Venezuelan isolate of *zucchini tigre mosaic virus* [34] suggesting that this residue might not have a direct effect on symptom development.

Cross-protection experiments under greenhouse conditions showed that PRSV-mild confers only a transient (90–120 days) protection against a secondary severe infection. Interestingly, isolate-specific PRSV detection after the 'protection' period failed to detect PRSV-mild; whereas PRSV-sev became readily detectable. In fact, field monitoring of infection dynamics showed that once mild-infected plants get superinfected by PRSV-sev, the latter prevails (Fig 4) resulting in the induction of severe symptoms, while the mild isolate is no longer detectable.

This phenomenon may be caused by non-homogeneous distribution of the mild isolate in the plant. Upon superinfection, the severe isolate colonizes non-infected cells at a faster rate, leading to symptom development and exclusion of the mild isolate. A similar phenomenon

was observed in cross-protection studies on *Cucurbita pepo* using PRSV-W strains, where timing of superinfection relative to the mild strain immunization, played a determinant role in the efficacy of the protection [22,38,39].

Superinfection exclusion (SIE), the basis for cross-protection, has been shown in different virus-plant systems [21]. In potyviruses, studies on *triticum mosaic virus* (TriMV) and *wheat streak mosaic virus* (WSMV), identified specific motifs in the NIa-Pro or CP which were SIE determinants [40]. In PRSV-mild, the NIa-Pro region was conserved across closely related isolates; whereas a two-amino acid deletion was observed in the CP. Whether this deletion has an impact on cell-to-cell and long-distance movement, and therefore SIE to PRSV-sev, requires further investigation.

TEM in leaf thin sections from PRSV-sev infected plants showed typical pinwheel inclusion bodies next to clusters of virions; however, these structures were not observed in leaf sections from samples infected with the PRSV-mild isolate, suggesting lower titers of the virus. Whether the KK/E deletion, has a direct impact on virion formation and/or accumulation also needs to be investigated.

Primers developed to detect, differentially, both isolates revealed that PRSV-mild is prevalent in commercial plantings from Los Ríos, the largest papaya production area in the country where the mild isolate was originally detected, and to a lesser extent in Santo Domingo, a neighboring province (Table 1). The geographical proximity between Los Ríos and Santo Domingo could explain the occurrence of the mild isolate only in these two provinces. In Santa Elena, Guayas and Manabí, where intensive papaya production was initiated more recently, PRSV-mild was not detected. From the epidemiological standpoint, these findings suggest that PRSV-mild may be dominant in Los Ríos province because severely affected trees are more commonly rogued. These apparently mild isolates will eventually cause symptoms and will not protect papaya plants against severe isolates. Therefore, decisions about roguing in apparently asymptomatic or mildly affected fields should be based upon lab-based PRSV diagnostics and not solely on visual inspections. In fact, relying on visual inspections alone to rogue infected plants favors the mild isolate remaining in the field longer to serve as a source of inoculum for the disease to spread.

Lastly, while PRSV-mild was only detected in the neighboring provinces Los Ríos and Santo Domingo, its occurrence in other provinces should not be discounted, because PRSV-sev appears to displace the mild isolate. Further studies are needed to asses more fully the extent of PRSV infections in Ecuadorian papaya production regions. To our knowledge, this is the first work to characterize a mild isolate of PRSV type-P from Ecuador, and to investigate its potential for use in cross-protection.

## Supporting information

**S1 Fig. Genome organization of papaya ringspot virus (PRSV).** A) Canonical genome organization of PRSV. B) A fraction of the coat protein (CP) alignment across related taxa, including sequences from Ecuador (Ec), Mexico (Mex), Hawaii (USA-HI), Texas (USA-Tx), Colombia (Col) and India (Ind). The aphid transmission determinant 'DAG' motif and the EK region are indicated, where the mild isolate from Ecuador, along with four additional isolates from the USA, Mexico and India, lack two amino acids. NCBI accession numbers are shown on the left.
(TIF)

**S2 Fig. Transmission electron microscopy (TEM) images.** A) The cytoplasm of leaf mesophyll cells infected with the severe isolate of papaya ringspot virus (PRSV-sev) showing pinwheel inclusions next to filamentous virus particles. B) The cytoplasm of leaf mesophyll cells

infected with the mild isolate of papaya ringspot virus (PRSV-mild) where no pinwheels inclusions or filamentous particles are observed.
(TIF)

**S3 Fig. Detection of the mild and severe isolate of papaya ringspot virus (PRSV-mild and PRSV-sev) by isolate-specific primers.** The size in base-pairs (bp) of each amplification product is shown. Sensitivity and specificity for each primer set was tested in single- and duplex. *In vitro* RNA dilutions started from an initial amount of $3 \times 10^6$ copies for each target (1ul) resuspended in 9 ul of virus-free papaya total RNA at a concentration of 100 ng/ul. Ten-fold dilutions followed using this target-total RNA mixture in molecular biology grade water and used as template (1ul) for reverse transcription in a 10 ul reaction. M: DNA 100bp ladder. (+) positive control (-) negative controls.
(TIF)

**S1 File.**
(JPG)

**S2 File.**
(JPG)

**S3 File.**
(JPG)

**S4 File.**
(JPG)

**S5 File.**
(JPG)

**S6 File.**
(JPG)

**S7 File.**
(TXT)

# Acknowledgments

The authors thank papaya growers in the five provinces surveyed in this study for granting access to their fields.

# Author Contributions

**Conceptualization:** Diego F. Quito-Avila.

**Data curation:** Sam Grinstead, Dimitre Mollov.

**Formal analysis:** Andres X. Medina-Salguero, Juan F. Cornejo-Franco, Diego F. Quito-Avila.

**Investigation:** Andres X. Medina-Salguero, Juan F. Cornejo-Franco, Diego F. Quito-Avila.

**Methodology:** Andres X. Medina-Salguero, Juan F. Cornejo-Franco, Joseph Mowery, Dimitre Mollov, Diego F. Quito-Avila.

**Resources:** Dimitre Mollov, Diego F. Quito-Avila.

**Writing – original draft:** Diego F. Quito-Avila.

**Writing – review & editing:** Dimitre Mollov, Diego F. Quito-Avila.

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
