## [Decision Letter · Decision Letter 0]

26 Nov 2020

PONE-D-20-32443

Genetic characterization of a mild isolate of papaya ringspot virus type-P (PRSV-P) and assessment of its cross-protection potential under greenhouse and field conditions

PLOS ONE

Dear Dr. Quito-Avila,

Thank you for submitting your manuscript to PLOS ONE. After careful consideration, we feel that it has merit but does not fully meet PLOS ONE’s publication criteria as it currently stands. Therefore, we invite you to submit a revised version of the manuscript that addresses the points raised during the review process.

We look forward to receiving your revised manuscript.

Kind regards,

Hanu R Pappu

Academic Editor

PLOS ONE

Journal Requirements:

2.We note that [Figure(s) S4] in your submission contain map images which may be copyrighted. All PLOS content is published under the Creative Commons Attribution License (CC BY 4.0), which means that the manuscript, images, and Supporting Information files will be freely available online, and any third party is permitted to access, download, copy, distribute, and use these materials in any way, even commercially, with proper attribution. For these reasons, we cannot publish previously copyrighted maps or satellite images created using proprietary data, such as Google software (Google Maps, Street View, and Earth). For more information, see our copyright guidelines: http://journals.plos.org/plosone/s/licenses-and-copyright.

1.    You may seek permission from the original copyright holder of Figure(s) [S4] to publish the content specifically under the CC BY 4.0 license. 

3. Your ethics statement should only appear in the Methods section of your manuscript. If your ethics statement is written in any section besides the Methods, please move it to the Methods section and delete it from any other section. Please ensure that your ethics statement is included in your manuscript, as the ethics statement entered into the online submission form will not be published alongside your manuscript

4.PLOS ONE now requires that authors provide the original uncropped and unadjusted images underlying all blot or gel results reported in a submission’s figures or Supporting Information files. This policy and the journal’s other requirements for blot/gel reporting and figure preparation are described in detail at https://journals.plos.org/plosone/s/figures#loc-blot-and-gel-reporting-requirements and https://journals.plos.org/plosone/s/figures#loc-preparing-figures-from-image-files. When you submit your revised manuscript, please ensure that your figures adhere fully to these guidelines and provide the original underlying images for all blot or gel data reported in your submission. See the following link for instructions on providing the original image data: https://journals.plos.org/plosone/s/figures#loc-original-images-for-blots-and-gels.

Reviewers' comments:

Reviewer's Responses to Questions

**Comments to the Author**

1. Is the manuscript technically sound, and do the data support the conclusions?

Reviewer #1: Partly

Reviewer #2: Yes

Reviewer #3: Partly

2. Has the statistical analysis been performed appropriately and rigorously? 

Reviewer #1: No

Reviewer #2: Yes

Reviewer #3: N/A

3. Have the authors made all data underlying the findings in their manuscript fully available?

Reviewer #1: Yes

Reviewer #2: Yes

Reviewer #3: Yes

4. Is the manuscript presented in an intelligible fashion and written in standard English?

Reviewer #1: Yes

Reviewer #2: Yes

Reviewer #3: Yes

5. Review Comments to the Author

Reviewer #1: The second objective of the study is how this mild isolate of PRSVE confer cross-protection against a severe isolate for which the experiment conducted are not sufficient enough to conclude that the isolate does not protect or if it does not protect, then the reasons given are not so sound.

In order to say that the mild virus vanished over a period when severe strain is challenge inoculated. Real time PCR based quantitation would have been a soundproof that the population of mild strain gradually came down and became undetectable. Mere PCR may give wrong results as failure in detection could have happen due to mutations or recombination in the mild strain.

When there is no cross protection, should it be called cross protected plants?

Whether five plants each variety inoculated with mild strain is enough to conclude the immunity or cross protection in PRSV? There should be quite a good number of plants with statistically designed experiment is needed to assess whether these sentinel plants are immunized? and then planted?

Why only one variety (12 plants) has been used in this border row (Sentinel plants) planting with healthy Tainung plants for the field experiment to study the cross protection potential of mild strain. Why these plants were not randomly planted inside the commercial plantation

Zucchini tigre mosaic virus is not a PRSV strain, it is a distinct species of potyvirus (Wang et al 2019). Does it infect papaya? Can we compare unrelated virus for concluding " no effect " with a minor change in FRNK-FRNR.

The comments and questions are given in the pdf copy of MS

Reviewer #2: Why qPCR was done to monitor the interaction of mild and severe in a single plant to assess the degradation process of mild one over severe one?

Is there any observation available that during the interaction, mild one become more virulent?

whether the variation observed in the symptom determinants incase of mild isolate was observed in all the samples collected from all place during the survey?

Reviewer #3: The manuscript concerns the molecular characterization of a mild isolate of PRSV and the evaluation of its possible use in cross-protection programs against a severe strain. Further, field surveys were conducted and persistence of the two strains in field conditions has been evaluated. In the course of the monitoring, no cross-protection effect was evident and the severe isolates revealed to be dominant in sentinel-plot experiments, in accordance to what observed during surveys, where the severe isolate was often detected. Apart from reporting the genome comparison, and evaluation of possible use for biological control, the manuscript does not provide significant novel information. Perceived novelty is indeed not the Journal focus for consideration for publication. However, the manuscript considers one mild strain against a severe strain, and in this sense a very limited situation. There might be many similar publications done in this exact same way, while I would support a more solid and general topic. The discussion contains indeed a series of consideration that are often speculative and that could otherwise be developed to answer some interesting questions, for example by generation of an infectious clone.

Please find below some comments:

L53: present in …..

L80: the authors state here “isolate from the same filed”.

L107: details on the isolates are still missing in the methods at this point

L111: which isolate was inoculated?

L126: Prime

L161: two-years

L173: two-months

L223: the number of obtained contigs is very high, 40.000: a different approach (host reads subtraction) and optimization of assembly parameters could results in better output; nevertheless, the genome could be assembled. This remains than a general comment, but the objective was achieved.

L230: number

L240: edit this sentence: the conserved….. were conserved…

L277: “instead” compared to what

L280-281: this sentence is not clear: in which conditions? In which host?

L401-407: these speculations, even indeed as such, are difficult to follow

L419-421: this comment on the protein motifs is not clear

L428: in their….

L433-434: what the authors can state is the evidence of absence of role of that aminoacid in the difference in symptoms between the isolates. However, the role of this aminoacid in the appearance of symptoms has been not molecularly investigated.

L454: only cell-to-cell? No long distance movement?

L475: few data report on the impact of the mild isolate in the field. Evaluations about epidemiology and roguing practicing appear also quite speculative.

6. PLOS authors have the option to publish the peer review history of their article (what does this mean?). If published, this will include your full peer review and any attached files.

Reviewer #1: **Yes: **RAMASAMY SELVARAJAN

Reviewer #2: **Yes: **Dr.T.Makeshkumar, ICAR-CTCRI, Thiruvananthapuram, india

Reviewer #3: No

---

## [Author Response · Author response to Decision Letter 0]

31 Dec 2020

Response to reviewers

Manuscript PONE-D-20-32443

The authors appreciate the comments and observations made by the three reviewers, as those will help improve the quality of the manuscript. We will address every comment and refer to the manuscript where corrections have been made. 

Review Comments to the Author

Reviewer #1: The second objective of the study is how this mild isolate of PRSVE confer cross-protection against a severe isolate for which the experiment conducted are not sufficient enough to conclude that the isolate does not protect or if it does not protect, then the reasons given are not so sound.

In order to say that the mild virus vanished over a period when severe strain is challenge inoculated. Real time PCR based quantitation would have been a soundproof that the population of mild strain gradually came down and became undetectable. Mere PCR may give wrong results as failure in detection could have happen due to mutations or recombination in the mild strain.

When there is no cross protection, should it be called cross protected plants?

Whether five plants each variety inoculated with mild strain is enough to conclude the immunity or cross protection in PRSV? There should be quite a good number of plants with statistically designed experiment is needed to assess whether these sentinel plants are immunized? and then planted?

Why only one variety (12 plants) has been used in this border row (Sentinel plants) planting with healthy Tainung plants for the field experiment to study the cross protection potential of mild strain. Why these plants were not randomly planted inside the commercial plantation

Zucchini tigre mosaic virus is not a PRSV strain, it is a distinct species of potyvirus (Wang et al 2019). Does it infect papaya? Can we compare unrelated virus for concluding " no effect " with a minor change in FRNK-FRNR.

The comments and questions are given in the pdf copy of MS

Response to reviewer #1:

The reason why real-time RT-PCR could not be used to monitor the mild and severe strains of PRSV was the little genetic variability observed between both isolates. Being 95.4% identical at the nucleotide level, the design of effective qPCR primers (including a probe) that satisfies the typical amplification size for a real-time PCR assay was not achieved during this study. Please realize that we’re not dealing with two different virus species; but two isolates of the same species. In fact, dozens of primers were tried and tested before achieving a clean specific amplification product for each isolate by conventional PCR, and it is our belief, that this effort should be appreciated.

As to the argument that “mere PCR may give wrong results as failure in detection could have happen due to mutations or recombination in the mild strain” the same argument could be applied for real-time detection (in the event that we would have achieved effective primers for the qPCR assay). The conventional RT-PCR assay was sensitive and specific enough to discriminate between both isolates. Despite not being able to monitor such a titer variation (by qPCR) in a given time frame, the fact that the transition from PRSV-mild to PRSV-sev infection was observed in two out of the twelve sentinel plants supports the usefulness of the conventional RT-PCR assay.

In regard to the reviewer’s concern “when there is no cross protection, should it be called cross protected plants?” We have been double-checked the context in which this phrase is used along the MS and have edited accordingly to comply with the reviewer suggestion.

We kindly disagree with the reviewer on his view on the lack of a good number of plants for the cross-protection greenhouse experiments. As long as the 5 -plant set (for each variety) includes a negative control, then it can be concluded that such an effect was attributed to the “pre-immunization” step. In addition, the field experiment showed a similar pattern, which brings us to the question about the sentinel plants. The sentinel plants were arranged randomly and not in a single row as understood by the reviewer. We are re-wording this part to make sure the reader understands exactly how this was done. Since we observed a similar response (transient cross-protecting effect) in both cultivars, and the field lot where we were allowed entrance for testing and experiments, we decided to keep consistency with the commercial cultivar ‘Tainung’. 

Finally, as to the Zucchini tigre mosaic virus. The reviewer is absolutely right. It is not a strain of PRSV but it is a potyvirus. As such, genome comparisons, especially when it comes to potential virulence or other biological factors, are warranted even across members of a same genus. 

Additional responses to comments added directly to the pdf:

Lines 112: 

Yes those are varieties. We have added that word to avoid confusion.

Line 133: 

We kindly disagree. Other studies have demonstrated that the removal of recombinant tracks before the phylogenetic analysis does not affect the topology of the tree; instead, it provides a global and better picture of the evolutionary pathway of the virus (please see Maina et al 2019. Genetic Connectivity Between Papaya Ringspot Virus Genomes from Papua New Guinea and Northern Australia, and New Recombination Insights).

185:

Sentinel plants were not immunized before planting. They were put out in the field as PRSV-free as stated in the methodology section.

239: 

Absolutely. That’s why we use the term ‘isolate’ instead of ‘strain’ to refer to the two variants of the virus.

271: 

Yes. We have clarified this.

278: 

Already responded above

283: 

This is mentioned in the discussion

290: 

Yes. Both isolates used for TEM were obtained from the same papaya variety

298: 

Fixed

303: 

No. we did not use semi- or quantitative ELISA.

307: 

We have added a sentence stating what moderate mosaic means under our study.

315: 

We have added a sentence stating that there were no mixed-infected samples during the field survey.

327: 

In the discussion, we mention about the apparent disappearance of the mild isolate after co-infection. 

386: 

This has been fixed

458: 

We have addressed the real time PCR issue above

Reviewer #2: Why qPCR was done to monitor the interaction of mild and severe in a single plant to assess the degradation process of mild one over severe one?

Is there any observation available that during the interaction, mild one become more virulent?

whether the variation observed in the symptom determinants in case of mild isolate was observed in all the samples collected from all place during the survey?

Response to reviewer #2:

Reviewer # 2 probably meant “why qPCR was NOT done?”. The same answer as above: The reason why real-time RT-PCR could not be used to monitor the mild and severe strains of PRSV was the little genetic variability observed between both isolates. Being 95.4% identical at the nucleotide level, the design of effective qPCR primers (including a probe) that satisfies the typical amplification size for a real-time PCR assay was not achieved during this study. 

Dozens of primers were tried and tested before achieving a clean specific amplification product for each isolate by conventional RT-PCR, and it is our belief, that this effort should be appreciated. In addition, the conventional RT-PCR assay was sensitive and specific enough to discriminate between both isolates. Despite not being able to monitor such a titer variation (by qPCR) in a given time frame, the fact that the transition from PRSV-mild to PRSV-sev infection was observed in two out of the twelve sentinel plants supports the usefulness of the conventional RT-PCR assay and the prevalence of the severe isolate over the mild isolate in mixed-infected plants, as observed in greenhouse experiments.

As stated in results, the mild isolate by itself exhibited a range of symptoms from subtle (mild) to some moderate mosaics; unlike the severe vein clearing and leaf deformation observed in plants infected either with the severe isolate only or both viruses (once the severe isolate prevailed in the infection).

Reviewer #3: The manuscript concerns the molecular characterization of a mild isolate of PRSV and the evaluation of its possible use in cross-protection programs against a severe strain. Further, field surveys were conducted and persistence of the two strains in field conditions has been evaluated. In the course of the monitoring, no cross-protection effect was evident and the severe isolates revealed to be dominant in sentinel-plot experiments, in accordance to what observed during surveys, where the severe isolate was often detected. 

Apart from reporting the genome comparison, and evaluation of possible use for biological control, the manuscript does not provide significant novel information. Perceived novelty is indeed not the Journal focus for consideration for publication. However, the manuscript considers one mild strain against a severe strain, and in this sense a very limited situation. There might be many similar publications done in this exact same way, while I would support a more solid and general topic. The discussion contains indeed a series of consideration that are often speculative and that could otherwise be developed to answer some interesting questions, for example by generation of an infectious clone.

First response to reviewer #3:

We kindly disagree with reviewer #3 on the lack of novelty of our work. Only a few studies have been documented on cross-protection in papaya. Perhaps the most remarkable of these studies was that conducted by Yeh and Gonsalves (1984) (Evaluation of induced mutants of papaya ringspot virus for control by cross protection. Phytopathology 74:1086-1091), where two artificially induced mild strains were obtained and tested in cross-protection. However, the protecting effect was not sustained under high disease pressure in the field. The authors then hypothesized that naturally occurring mild strains of PRSV could potentially act as cross-protection inducers in a more stable fashion.

Our work is one of a few studies where a naturally occurring isolate is assessed as cross-protection inducer. Even though our results were somehow similar to those reported by Yea and Gonsalves (transient protection), the importance and implications of the occurrence of this type of strains are presented and discussed in our work.

We agree on the infectious clone suggestion when it comes to elucidating the two amino acid deletion in the coat protein. However, that subject is not part of the present work and will be considered in the future.

Responses to additional comments:

L53: present in …..

We have reworded this sentence.

L80: the authors state here “isolate from the same filed”.

Yes, that is correct. We obtained both isolates from the same commercial field.

L107: details on the isolates are still missing in the methods at this point

Those details are written in the ‘virus source and testing’ section, specifically lines 88 – 90.

L111: which isolate was inoculated?

We thank the reviewer for catching this, as is indeed incomplete. We have modified that sentence to be more specific about the inoculated isolates.

L126: Prime

Fixed

L161: two-years

Two-year-old plantings should be grammatically correct

L173: two-months

Two-month-old papaya seedlings should be grammatically correct

L223: the number of obtained contigs is very high, 40.000: a different approach (host reads subtraction) and optimization of assembly parameters could results in better output; nevertheless, the genome could be assembled. This remains than a general comment, but the objective was achieved.

Yes. We appreciate the comment and suggestion. The genome was assembled without major issues.

L230: number

Fixed

L240: edit this sentence: the conserved….. were conserved…

Thanks for catching this one!.. The sentence has been reworded

L277: “instead” compared to what

We have modified this paragraph in order to make it clearer.

L280-281: this sentence is not clear: in which conditions? In which host?

We have modified this paragraph in order to make it clearer.

L401-407: these speculations, even indeed as such, are difficult to follow

Slightly modified.

L419-421: this comment on the protein motifs is not clear

Slightly modified.

L428: in their….

Fixed

L433-434: what the authors can state is the evidence of absence of role of that aminoacid in the difference in symptoms between the isolates. However, the role of this aminoacid in the appearance of symptoms has been not molecularly investigated.

We totally agree with this comment.

L454: only cell-to-cell? No long distance movement?

Yes. Long-distance movement as well. We added that phrase.

L475: few data report on the impact of the mild isolate in the field. Evaluations about epidemiology and roguing practicing appear also quite speculative.

We kindly disagree. There’s a reasonable correlation between the occurrence of PRSV-mild in provinces where papaya has been grown (using rouging as a major management strategy) for much longer periods compared to the occurrence in those with recent papaya production history. It is our belief that this should be part of the discussion.

---

## [Editor Report · Decision Letter 1]

21 Jan 2021

Genetic characterization of a mild isolate of papaya ringspot virus type-P (PRSV-P) and assessment of its cross-protection potential under greenhouse and field conditions

PONE-D-20-32443R1

Dear Dr. Quito-Avila,

We’re pleased to inform you that your manuscript has been judged scientifically suitable for publication and will be formally accepted for publication once it meets all outstanding technical requirements.

Kind regards,

Hanu R Pappu

Academic Editor

PLOS ONE
---

## [Editor Report · Acceptance letter]

25 Jan 2021

PONE-D-20-32443R1 

Genetic characterization of a mild isolate of papaya ringspot virus type-P (PRSV-P) and assessment of its cross-protection potential under greenhouse and field conditions 

Dear Dr. Quito-Avila:

I'm pleased to inform you that your manuscript has been deemed suitable for publication in PLOS ONE. Congratulations! Your manuscript is now with our production department. 

Kind regards, 

on behalf of

Dr. Hanu R Pappu 

Academic Editor

PLOS ONE